# Rumination Time, Reticulorumen Temperature, and Activity in Relation to Postpartum Health Status in Dairy Cows During Heat Stress

**DOI:** 10.3390/ani15111616

**Published:** 2025-05-30

**Authors:** Szilvia Szalai, Ákos Bodnár, Hedvig Fébel, Mikolt Bakony, Viktor Jurkovich

**Affiliations:** 1Department of Animal Technology and Animal Welfare, Institute of Animal Sciences, Hungarian University of Agriculture and Life Sciences, 2100 Gödöllő, Hungary; 2Department of Obstetrics and Food Animal Medicine Clinic, University of Veterinary Medicine, 1078 Budapest, Hungary; febel.hedvig@univet.hu; 3Centre for Translational Medicine, Semmelweis University, 1085 Budapest, Hungary; bakony.mikolt@semmelweis.hu; 4Centre for Animal Welfare, University of Veterinary Medicine, 1078 Budapest, Hungary; jurkovich.viktor@univet.hu

**Keywords:** calving, body temperature, precision livestock farming

## Abstract

This study explores how heat stress affects dairy cows around calving and how modern monitoring tools can help detect health issues early. Calving is a stressful time for cows, impacting their health, milk production, and overall well-being. This research monitored 40 cows before and after calving, tracking their body temperature, rumination, and activity. The goal was to see if these factors differed between healthy cows (*n* = 26) and those that developed illnesses (*n* = 14) after giving birth. The findings showed that all cows experienced heat stress. Rumination time decreased before calving, hitting the lowest point two days after birth in sick cows. Activity levels increased before calving but took longer to return to normal in sick cows. Therefore, this study highlights the importance of precision farming tools like rumen sensors, which can detect early signs of health problems, allowing farmers to intervene sooner. These insights can help farmers reduce disease risks and enhance milk production, leading to more sustainable dairy farming.

## 1. Introduction

The early detection of signs of disease is crucial to successful health management, particularly with efficient health monitoring during the transition period, significantly impacting longevity and lactation profitability [1]. Most health disorders occur within the first 30 days after calving, primarily due to endocrine changes and negative energy balance, which can lead to immune suppression [2,3]. A worldwide trend has emerged in recent years, with the average herd size consistently increasing, leading to a reduction in labor per cow [4]. Therefore, precision livestock farming tools, such as rumen boluses and neck tags, can be used to monitor the clinical parameters of animals, ensuring herd health [5,6].

Heat stress is a leading environmental stress factor for dairy cows [7,8]. The risk of heat stress for dairy cows is increasing primarily due to global warming and the forecasted growing number of heat stress days [9,10,11]. Heat stress in late gestation impairs immune function, raising the risk of mastitis and retained placenta [12]. It also weakens gut integrity, increases susceptibility to metabolic disorders [13], and affects hepatic response, gestation length, and calf weight [14]. Late pregnancy heat stress reduces colostrum quality, including IgG levels, due to nutritional restriction and decreased mammary blood flow [15], negatively impacting immune function in the dam and offspring and increasing morbidity and mortality [16,17,18]. Since the activity of the cow increases physiologically before calving [19], deviation in activity around calving (either increased or reduced activity) is indicative of additional pain during the calving process [20]. Heat stress may lead to increased activity in sick cows around calving, but this depends on several factors, including the severity of the illness, the cow’s physiological response, and environmental conditions. Heat stress may exacerbate discomfort in cows already experiencing pain and stress from calving complications (e.g., dystocia, milk fever, or retained placenta), leading to more movement as the cow struggles to cope [20,21].

Reticulorumen temperature (RRT) depends on the balance between heat production and loss [22]. Bacterial fermentation in the rumen generates extra heat, but cows reduce their dry matter intake in hot conditions to minimize heat production and maintain homeostasis [23]. The core body temperature ranges from 38.0 to 38.5 °C, with the rumen temperature typically exceeding this by approximately 1 °C, regardless of environmental heat [24]. Under conditions of thermal stress, when the temperature–humidity index (THI) is above 65, the reticulorumen temperature is raised [25], and elevated rumen temperatures disrupt microbial populations, metabolism, and immunity [26,27]. Reticulorumen boluses enable individual cow monitoring for preventive health measures [28,29].

Traditional rumination monitoring is labor intensive and inaccurate [30,31,32], but automated technologies are now widely used on dairy farms to measure rumination time (RT) [6,33]. Healthy cows ruminate on average 463 to 522 min daily [34]; however, pain, hunger, anxiety, or illness reduces rumination time [34,35]. Calving also affects RT, as it declines 4 h before and 8 h after parturition [36], with prolonged reductions in difficult calvings [37]. RT normalizes within 7–15 days postpartum, making early monitoring useful in detecting disease risk [38]. RT is a key welfare indicator [39] as it declines further during heat stress [21,40].

Deviation from normal daily activity patterns can indicate painful conditions and diseases in cows. Illness-related behaviors include lethargy, reduced social interaction, and a decrease in general physical activity, among other symptoms. However, reduced activity and increased lying time benefit energy conservation and maintain the immune response [41]. Individual, environmental, housing system, and management factors can influence daily activity patterns.

We hypothesized a difference in the reticulorumen temperature, the time spent ruminating, and the overall activity of healthy dairy cows and those with diseases postpartum on hot summer days. Our study, therefore, aimed to measure and compare the rumen and behavioral parameters between healthy and sick cows.

## 2. Materials and Methods

### 2.1. Animals and Farm Management

This study processed data from the PLF data loggers on a farm in Hungary only. The animals were not manipulated or sampled and were not subjected to pain or suffering. Considering this, this study is exempt from obtaining a permit under Act 28 of 1998 on the Protection and Welfare of Animals and Government Decree 40/2013 on Animal Experiments.

The study was conducted during the summer season on a large-scale dairy farm in Orosháza (46°35′26.945″ N, 20°38′39.582″ E), which housed approximately 500 Holstein Friesian dairy cows and their offspring. Forty clinically healthy, multiparous cows (parity: 3.1 ± 1.2; body condition score: 3.7 ± 0.3; average milk yield in previous 305 d lactation: 9197 ± 2036 kg) were enrolled in the study two weeks before their expected calving. The cows were not lame (assessed according to [42]) and had no other clinical issues (e.g., mastitis) before or during enrollment. The cows were kept in the calving pen, bedded with straw, and the feed bunk was on the outer runway. Total mixed ratio (TMR; Table 1 and Table 2) was delivered once a day in the morning and pushed back regularly. Drinking water was provided ad libitum during the study.

The cows were monitored hourly for signs of calving, and interventions were performed only in dystocia cases. The cows were milked within one hour after calving. From day 0 to 3 postpartum, cows stayed in a separate straw-bedded barn for freshly calved cows and fed and milked twice daily. The farm veterinarian checked the cows’ health daily, and the diseased animals were treated according to their specific problems and the farm protocol. Then, the animals were moved to the main lactating herd in a straw-bedded, free-stall barn. The same TMR as fresh cows was delivered twice a day. Cows were milked twice daily and monitored throughout the transition period. The study groups were formed retrospectively according to the peripartum health status of the cows, based on the farm vet’s diagnosis. A cow was considered healthy and allocated to the healthy (HE) group (*n* = 26) if she was not affected by any postpartum health disorders until the end of the study period (14 days in milk [DIM]). A cow was considered diseased and allocated to the diseased group (DI; *n* = 14) if she had been diagnosed with mastitis, metritis, lameness, or ketosis by the farm veterinarian up to 14 DIM.

### 2.2. Barn Microclimate Measurements

Weather data loggers (Voltcraft DL-181 THP; Conrad Electronic SE, Hirschau, Germany) were placed above the cows in the barns from the end of June, and data were recorded until the end of August to measure the microclimate. The tool recorded the dry air temperature and relative humidity hourly. From the measured values, the temperature–humidity index (*THI*) was calculated by the following formula [43]:$$THI=\left(0.8\times T\right)+\left(\frac{RH}{100}\right)\times \left(T-14.4\right)+46.6$$
where *T* stands for dry air temperature (°C), *RH* stands for relative humidity (%). The *THI* threshold value for heat stress was 68 [11,44,45,46].

### 2.3. Recording of Rumination Time and Activity

Rumination time was recorded from 5 days before calving to 14 days in milk (DIM), using a microphone-based rumination and activity monitoring sensor (Ruminact HR-Tags, SCR Engineers Ltd., Netanya, Israel). The transponder was placed on the left side of each cow’s neck. The sensor identified rumination time (RT) by detecting the sound of regurgitation and specific sound patterns related to rumination behavior and tracking overall movement intensity [47,48,49]. The collar used a built-in microphone to record and identify regurgitation and rumination. As defined by the software, a rumination event began when the system detected the sound associated with regurgitation and expressed it as rumination time in minutes [47,50]. The data were saved to a microcell every 2 h; the 12 microcells allowed 24 h of data storage.

A cow’s activity was translated into an index value ranging from 0 to 255 [49], representing weighted standard deviations from each cow’s baseline activity. Data were exported from the Heatime Pro software and converted into spreadsheet files for further analysis.

### 2.4. Reticulorumen Temperature

An indwelling wireless rumen bolus (SmaXtec Animal Care GmbH, Graz, Austria) was used to monitor reticulorumen temperature from 5 days before calving to 14 DIM. After matching the bolus and ear tag numbers, the boluses were administered orally using a special bolus gun. The measurement interval was 10 min, and the generated data were immediately sent to the cloud (SmaXtec Cloud) via an internet connection by a reader device. The measured parameters were displayed on various graphs through the SmaXtec Application (v. 1.9.1), and the data were downloaded into spreadsheet software for further analysis. Data obtained from the SmaXtec bolus included rumen temperature (“temp_without_drink_cycles”), the raw rumen temperature corrected by the manufacturer’s proprietary algorithm for the effects of drinking events [51].

### 2.5. Statistical Analysis

The data visualization and analysis were performed in the R statistical environment (version 4.3.2) [52], using the basic functions as well as the ‘ggplot2’ (v. 3.5.2), ‘rmcorr’ (v. 0.7.0), ‘nlme’ (v. 3.1-168), ‘epitools’ (v. 0.5-10.1), and ‘emmeans’ (v. 1.11.1) packages. From the large number of daily temperature measurements, summary indicators for each day (daily average and maximum rumen temperature, environmental temperature, and THI) were calculated first. The rumination time was summed up for each day, representing the daily rumination time. These daily indicators were used as input for the analysis. It was also recorded whether metabolic diseases, lameness, uterine infections, or mastitis were diagnosed in the animals during the postpartum period, which was also used as a predictor during analysis.

The relationship between the average and maximum daily rumen temperature and the average and maximum daily THI was characterized by the correlation coefficient. As repeated measurements were taken on the same animal, the observations were not independent, which meant that Pearson’s or Spearman’s coefficients could not be calculated. We followed the method proposed by Bland and Altman [53,54], which provided a correlation coefficient for repeated measurements (ranging from −1 to 1).

The mean daily rumination time and activity were compared between groups using a linear mixed-effect model. Mixed models term predictors as ‘fixed’ effects and take the non-independence of repeated measures on the same subject into account by including a ‘random’ effect for each subject in the covariance matrix, thereby differentiating within-subject, between-subject, and sampling variance.

The response variables were daily rumination time and raw activity units as measured by the SCH Heatime system. The fixed effects were days concerning calving, postpartum health status (healthy or diseased), and average THI (included as a controlling variable). We used the following model formula for the calculations:$$\begin{array}{c} Y_{ijkl}=intercept+{ß}_{i}*DIM+{ß}_{j}*postpartum\;health\;status\;+\\ {ß}_{k}*average\;THI+\eta_{l}+\varepsilon_{ijkl} \end{array}$$
where intercept represents least squares mean of *Y* in the reference level of days in relation to calving, the reference level of postpartum health status, and the reference level of average *THI*; ß(*i*) represents the change in *Y* in i-th level of days in relation to calving; ß(*j*) represents the change in *Y* in j-th level of postpartum health status; ß(*k*) represents the change in *Y* per unit of increase in *THI*; *η(l)* adds a random term to the intercept for the l-th cow; *ε(ijkl)* represents random measurement error.

## 3. Results

### 3.1. Barn Microclimate and Animal Temperature

Based on the daily THI data, continuous heat stress was observed during the study period, with values consistently above the 68 THI threshold, peaking at around 80 THI, with minimal variability (Figure 1). The rumen temperature ranged between 39 °C and 41 °C, indicating that the animals were experiencing heat stress.

The correlation coefficient between the maximum daily rumen temperatures and the maximum daily THI was 0.27 (95% CI: 0.20; 0.33, *p* < 0.0001; Figure 2). Based on the low magnitude of the correlation coefficients, the rumen temperature shows a significant but moderate correlation with the temperature–humidity index.

Figure 3 shows that the average reticulorumen temperature exhibits a similar pattern in healthy and diseased cows around calving, with a nadir on the day of calving. The daily average rumen temperature showed a significant correlation with the number of days remaining until calving and the number of days since calving (*p* < 0.0001), as well as with the THI (*p* < 0.0001). However, it was not significantly dependent on the presence of postpartum diseases (*p* = 0.463) during the study period (−5 to 14 DIM).

### 3.2. Rumination Time Around Calving

Figure 4 displays the pattern of rumination time in the two study groups around calving (from −5 to 14 DIM). Rumination time significantly changed around the time of calving. In this study setup, the effect of heat stress conditions on the relationship between the time spent ruminating and the number of days remaining until calving, as well as the number of days since calving, was examined. It was also investigated whether the need for postpartum treatment could be predicted in any way from the pattern of temporal changes in rumination time. Daily rumination time decreased throughout the prepartum period, with the lowest value observed on the day before calving in the HE group, and reached its nadir on 2 DIM in the DI group.

The average daily rumination time was found to be significantly influenced by the number of days around calving (*p* < 0.0001), the presence of postpartum problems (*p* < 0.0001), and the THI (*p* = 0.029).

There was no significant difference between the two groups just before and after calving; it was assumed that this was due to the differing degrees of change in rumination time between them before calving. Therefore, we analyzed this close peripartum period (−1 to 2 DIM, indicated with a blue background in Figure 5). Figure 6 displays the results of this separate analysis, showing a significant difference between the two groups on days −1 and 2. It is also evident in Figure 5 that RT decreased in the HE group, whereas it remained unchanged in the DI group around calving.

### 3.3. Activity Changes Before and After Calving

Figure 6 displays the activity of the cows in the study period. The activity increased before calving, and there was no difference between the two groups. Activity returned to its normal level at 4 DIM in HE cows. Diseased cows’ activity was significantly higher than that of the HE cows at 4 DIM, and it returned to the normal level at 5–7 DIM. From this time, there was no difference between the two groups.

## 4. Discussion

Calving is a challenging period for dairy cows, involving physiological changes and environmental stressors. DMI decreases by around 30% on the calving day [47], and rumination time drops by 70% compared with the dry period [38,55]. Fetal nutrient demand peaks three weeks pre-calving, and heat stress during the dry period reduces milk production and affects hepatic metabolism [56]. Metabolic disorders in early lactation often lead to further health issues, underscoring the importance of optimal nutrition and welfare [57]. Ketosis, associated with NEB, occurs before and after calving, leading to inflammation and organ damage [58]. Heat stress-induced feed intake depression worsens NEB and immune dysfunction, increasing disease risks, such as retained placenta, metritis, mastitis, and lameness [46,59,60]. The symptoms mentioned above also occurred in our study, with 14 (35%) of the 40 selected animals becoming ill after calving.

Reticulorumen temperature decreases typically shortly before parturition, with reductions ranging from 0.4 to 1.0 °C resulting from both hormonal and behavioral alterations [61]. Rumen temperature is influenced by the balance between internal heat production and heat dissipation to the environment [22]. Additionally, cattle exhibit seasonal body temperature fluctuations, corresponding to ambient environmental conditions [62]. Even a modest rise in body temperature by 1 °C or less could impair performance in dairy cows, making body temperature a sensitive indicator of physiological responses to heat stress, as it remains relatively stable under normal conditions [63]. Studies have shown that body temperature decreases on the day of calving, which may allow the prediction of parturition within a 24 h window [55,64]. However, individual variability in temperature responses can limit the predictive accuracy of temperature alone [65]. In our study, changes in RRT appeared to be more strongly associated with the calving process than with the THI during the periparturient period. Although no statistically significant differences were observed between the two groups, we propose that RRT may serve as a valuable predictor of calving, not only under thermoneutral conditions but also during periods of heat stress, as a distinct drop in RRT was consistently observed before parturition, even on the hottest days of the year.

Sensor-based monitoring systems enable the detection of alterations that may indicate sickness or disease risk [66]. Cows with reduced rumination time before calving are more likely to develop health disorders during the peripartum phase (metabolic disorders, mastitis, and metritis) than cows with a greater rumination time antepartum [67]. A slower increase in rumination time after calving is associated with severe inflammation around parturition. Kovács et al. [55] found that cows with dystocia had lower rumination times (within 8 h before parturition) than cows with normal calving and remained depressed for extended periods. Similar patterns were observed in our study, where animals were divided into two groups based on their health status after parturition: healthy and diseased cows. The cows that became sick during the transition already had lower rumination times before calving and took longer to return to their pre-parturition levels. Soriani et al. [34] determined the average rumination time to be 522 min daily in a stress-free environment for healthy multiparous dairy cows. Of course, various factors, such as diet, management, lactation stage, or environmental conditions (such as heat stress), can influence RT [35]. The fact that rumination activity is influenced by acute stress [68] suggests that cows with dystocia or any health disorders may experience a higher stress level as parturition approaches. Similar to our results, other researchers found a significant negative correlation between RT and THI, with RT decreasing as THI increases [39,40,69]. Paudyal et al. [70] also demonstrated the effect of season on RT around calving, as cows with health disorders (NEB, ketosis, dystocia, etc.) during the hot season had a lower average daily RT than healthy cows. Reticulorumen temperature decreases shortly before parturition, ranging from 0.4 to 1.0 °C, and is attributed to endocrine and behavioral changes [61]. We showed a similar pattern in our study, and the change in RRT appeared to be more influenced by calving than by changes in THI during this period.

Most studies agree, as we also showed, that 24 h before calving, cows with dystocia and other health disorders, such as ketosis, mastitis, and metritis, exhibit increased restlessness compared to eutotic or healthy cows [28,71,72]. However, restlessness encompasses standing and lying bouts, walking, and weight distribution, all indicating discomfort and pain in animals [71]. Heat-stressed cows spend more time standing to increase their body surface area for heat dissipation, but it may disrupt the calving process and lead to more difficulties during labor. Similarly, our results are consistent with those of other studies, which have shown an increase in activity as calving approaches [73]; however, this increase can depend on the health status of the animals. However, according to Antanaitis et al. [28], the activity of sick cows 12 days before calving increases sharply, especially in cows likely to experience milk fever and placenta retention after parturition. In contrast, the activity decreases in healthy cows and those with clinical mastitis shortly after calving. We could not create additional subgroups within the DI group due to the limited number of animals and because most cows had multiple diseases. Stress caused by high ambient temperatures worsens the health status of the animal by increasing panting and standing behavior, while reducing rumination, which can delay recovery.

In this study, we were curious to see if the measured parameters before calving in otherwise healthy cows could detect changes to help predict which ones would become sick after calving. Such a parameter seems to exist, but our sample size for making the prediction was small, which is an explicit limitation of our study. This calculation could be performed in future studies with a larger dataset.

## 5. Conclusions

Heat stress during the periparturient period significantly impacts dairy cow health by altering rumination time, reticulorumen temperature, and activity. Our study confirmed that cows developing postpartum diseases already exhibited reduced rumination time in the prepartum phase. The reticulorumen temperature dropped around calving, influenced more by parturition than heat stress, although a moderate correlation with THI was observed. Activity increased before calving but remained elevated longer in diseased cows, likely due to discomfort and anxiety. These findings highlight the potential of these indicators and the importance of PLF tools to predict diseases. Further studies on a larger sample are necessary to develop appropriate algorithms for predicting diseases in the transition period.

## Figures and Tables

**Figure 1 animals-15-01616-f001:**
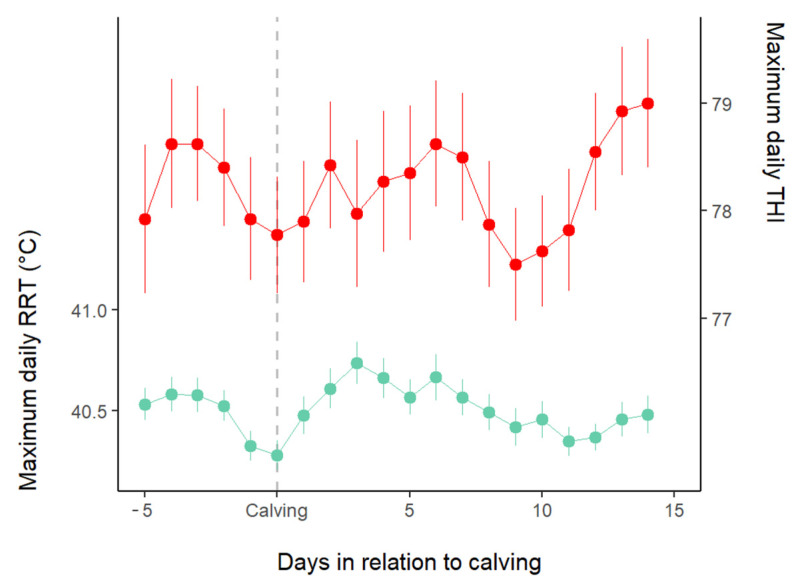
Average daily maximum reticulorumen temperature (RRT; mean ± SE; green line) of all cows and daily maximum temperature–humidity index (THI; red line) around parturition.

**Figure 2 animals-15-01616-f002:**
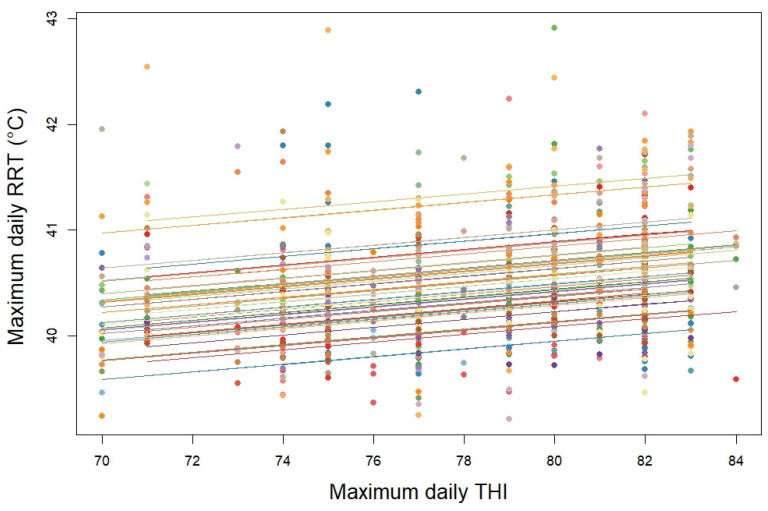
Correlation between daily maximal reticulorumen temperature (RRT) and temperature–humidity index (THI). The colored lines represent the relationship for each cow. The dots in different colors represent the RRT values of each cow at the given THI value.

**Figure 3 animals-15-01616-f003:**
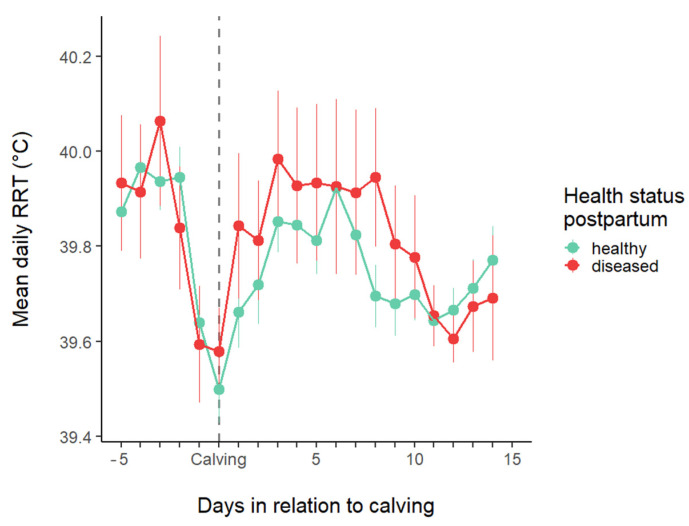
The average daily reticulorumen temperature (RRT; mean ± SE) around parturition.

**Figure 4 animals-15-01616-f004:**
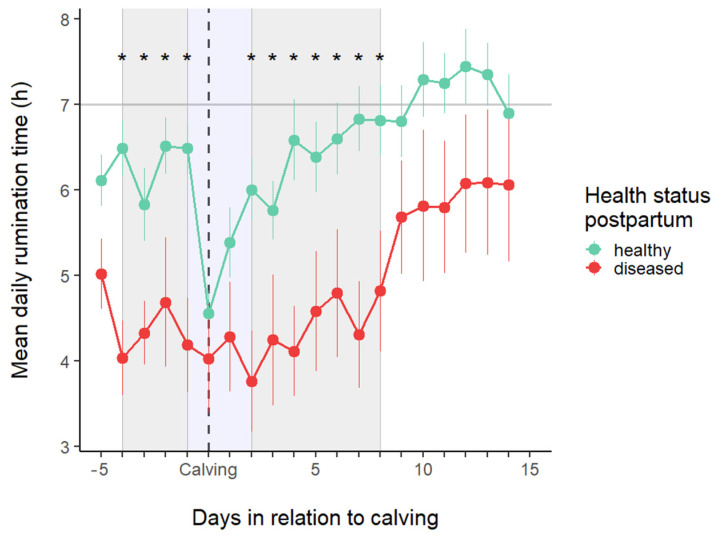
Average rumination (mean ± SE) time in healthy and diseased cows during the study period. Asterisks indicate the days where a significant difference (*p* < 0.05) exists between groups. These periods are also marked with a gray background.

**Figure 5 animals-15-01616-f005:**
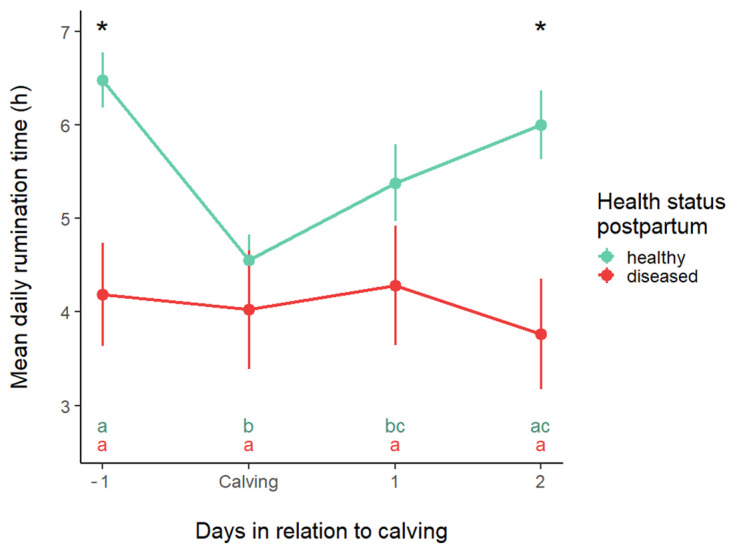
Average rumination time (mean ± SE) in healthy and sick cows in the close peripartum period. Asterisks indicate the differences between the groups; the letters represent the differences between the days within each group. If none of the letters match between the labels of two days, the difference between their averages is significant.

**Figure 6 animals-15-01616-f006:**
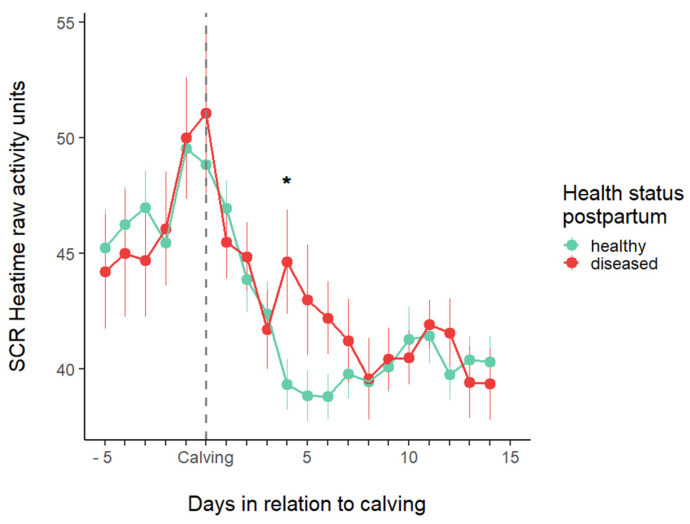
Average daily activity (mean ± SE) in the study period. The asterisk indicates a significant difference (*p* < 0.05) between the two groups.

**Table 1 animals-15-01616-t001:** Ingredients of the TMR before and after calving.

Ingredients (kg)	Before Calving	After Calving
Corn silage	8.0	9.0
Italian ryegrass haylage	3.0	10.0
Wheat straw	3.0	0.0
Wet corn gluten feed	4.0	6.5
Concentrates	4.3	13.0
In total	22.3	38.5

**Table 2 animals-15-01616-t002:** Calculated chemical composition of the TMR before and after calving.

Items	Before Calving	After Calving
Dry matter (%)	52.7	52.5
NE_l_ (MJ/kgDM)	5.6	7.3
Crude protein (%DM)	15.9	17.3
Sugar (%DM)	3.8	9.9
Starch (%DM)	16.5	21.0
Crude fiber (%DM)	20.0	13.1
ADF (%DM)	23.3	17.9
NDF (%DM)	34.0	31.3
NFC (%DM)	38.1	38.1
Ether extract (%DM)	3.6	4.3
Ash (%DM)	8.3	8.9
Ca (%DM)	1.7	1.1
P (%DM)	0.4	0.5

## Data Availability

The data presented in this study are available upon request from the corresponding authors.

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
