# Peer review of "Rumination Time, Reticulorumen Temperature, and Activity in Relation to Postpartum Health Status in Dairy Cows During Heat Stress"

_animals, 2025, doi:10.3390/ani15111616_

Round 1
Reviewer 1 Report
Comments and Suggestions for Authors
Peer Review Summary – Manuscript ID: animals-3590616
Title: Assessing the impact of heat stress on rumination time, reticulorumen temperature, and activity during the periparturient period in dairy cows
The manuscript presents relevant and timely research within the scope of Animals, addressing the impact of heat stress on physiological and behavioral parameters in dairy cows during the periparturient period. The title is appropriate, and the abstract effectively summarizes the main findings. The Simple Summary begins well but requires improvements from line 24 onward for better clarity and flow.
The introduction presents a clear objective and hypothesis. However, it would benefit from additional references to support key statements, as well as some adjustments in the use of abbreviations and paragraph organization.
The Materials and Methods section needs clarification regarding the exemption from animal ethics committee approval, which should be explicitly stated. Additionally, the THI model used should be updated, and I recommend referring to Berman et al. (2016) for a more robust model https://doi.org/10.1007/s00484-016-1136-9.
The statistical analyses need to be more clearly described, including the significance level adopted.
Results are well presented and relevant to the study objectives. The discussion is appropriate but could briefly mention climate change as a factor that may exacerbate heat stress effects in the future.
The conclusion should be revised to remove lines 311–314 and rewritten to be more objective and aligned with the study's findings.
Lastly, reference formatting should be reviewed—see, for example, reference 34.
I am sending a PDF file with specific corrections and suggestions.

Author Response
Rev.: The manuscript presents relevant and timely research within the scope of Animals, addressing the impact of heat stress on physiological and behavioral parameters in dairy cows during the periparturient period. The title is appropriate, and the abstract effectively summarizes the main findings. The Simple Summary begins well but requires improvements from line 24 onward for better clarity and flow.
AU: Thank you for your valuable comments and suggestions. We tried to clarify the text accordingly.
Rev.: The introduction presents a clear objective and hypothesis. However, it would benefit from additional references to support key statements, as well as some adjustments in the use of abbreviations and paragraph organization.
AU: Thank you. We adjusted the text according to your suggestions indicated in your PDF file.
Rev.: The Materials and Methods section needs clarification regarding the exemption from animal ethics committee approval, which should be explicitly stated. Additionally, the THI model used should be updated, and I recommend referring to Berman et al. (2016) for a more robust model https://doi.org/10.1007/s00484-016-1136-9.
AU: Thank you for these suggestions. They were taken into account.
We have calculated the THIs according to the Berman et al. (2016) formula and measured the correlation between the values obtained with the two formulas. The Pearson correlation coefficient was 0.9987 for average THIs and 0.9980 for maximum THIs. The average THIs were 0.09 units higher when calculated with the Berman (2016) formula, and maximum THIs were, on average, 3.27 units higher when calculated with the Berman (2016) formula, in comparison to the Mader (2006) formula. We concluded that the strength of correlation and direction of numerical differences would not change our conclusions on the level of heat stress, therefore, we do not wish to update the calculations.
Rev.: The statistical analyses need to be more clearly described, including the significance level adopted.
AU: Thank you for your suggestion. The conditions of applicability for calculating Pearson and Spearman correlations include independence of observations. This in our study did not hold, as several measurements were taken on the same animals. It necessitated using a correlation measure to separate within-subject variability from between-subject variability.
Rev.: Results are well presented and relevant to the study objectives. The discussion is appropriate but could briefly mention climate change as a factor that may exacerbate heat stress effects in the future.
AU: Thank you for pointing this out. We mentioned it in the introduction.
Rev.: The conclusion should be revised to remove lines 311–314 and rewritten to be more objective and aligned with the study's findings.
AU: The conclusion has been rephrased.
Rev.: Lastly, reference formatting should be reviewed—see, for example, reference 34.
AU: References were checked for formatting.
Rev.: I am sending a PDF file with specific corrections and suggestions.
AU: Thank you. We are grateful for your suggestions.
Reviewer 2 Report
Comments and Suggestions for Authors
The paper, titled “Assessing the impact of heat stress on rumination time, reticulorumen temperature, and activity during the periparturient period in dairy cows” addresses an important and timely topic in animal science. I found the subject matter of the article absolutley fascinating, and read the manuscript with great interest. The paper aligns well with the scope of the journal, falling as it does into a very relevant area of study. However, I do believe that in its current form, it has several shortcomings that need to be addressed before it can be considered for publication. These issues, which I detail below, range from minor presentational things to some more significant methodological concerns, unfortunatly.
Introduction
- It provides a well overview of the studys context, but it could benefit from clearer articulation of the specific knowledge gap being addressed. But the authors report the importance of early disease detection/identification and the use of PLF tools, the introduction doesn't explicitly state what is lacking in the present literature that this study aims to fill the gap. It would be great to see a sentence or two highlighting what new insights this research is expected to provide. See the suggestions:
- Line 58: I suggest citing 10.3389/fvets.2023.1141286 to include more information regarding the longevity and profitability.
- Line 60: I suggest citing 10.3389/fvets.2024.1437352 to include more information regarding the metabolic disease.
- Line 68: I suggest citing 10.1016/j.vas.2024.100363 to include more information regarding heat stress.
- Line 72: I suggest citing 10.3390/ani12162129 regarding the monitoring using boluses for not only temperature but also pH.
- Line 81: I suggest citing 10.3390/ani15030458 and 10.3389/fanim.2025.1547395 regarding PLF technologies to monitor rumination time.
- References reported is quite comprehensive, but at times it feels a bit like a list of facts. To improve the flow and impact, the authors could try to synthesize the existing research more, perhaps by grouping studies that address similar aspects of the topic and by highlighting any conflicting findings or areas of uncertainty. This would help to improve a stronger narrative and make the research question stand out more.
- it mentions many factors that influence RT and activity, such as heat stress, calving, and health disorders. but, the relative importance of these factors and how they might interact isn't always clear. It can be helpful if the authors could provide a bit more context, perhaps by explaining which factors are considered most critical in the peri-parturient period and how this study will help to disentangle their effects.
- "absolutley" should be "absolutely"
- "unfortunatly" should be "unfortunately"
- There are a few instances where the phrasing is a bit informal for a scientific paper. For example, "it feels a bit like a list of facts." Consider rephrasing these for a more academic tone.
Materials and Methods
- The description of the used animal selection and grouping is generally good, but I have a question about the criteria for defining "clinically healthy" cows as including factor. It would be usefull to have a bit more details on what specific health parameters were assessed and what thresholds were used to classify dairy cows as healthy. Also, were there specific cut-offs used for BCS? Report references thanks
- The section on barn microclimate measurements explains how the THI was calculated, which is good. However, it might be worth adding a brief justification for why the THI threshold of 68 was chosen. Is this a widely accepted value in the literature, or is there something specific to this study that warrants its use?
- In the description of the rumination time and activity monitoring, it would be helpful to provide a bit more detail on how the Ruminact HR-Tags work. While the authors mention that the sensor detects the sound of regurgitation, it would be interesting to know more about the specifics of the technology. For example, how accurate and reliable is it? Are there any limitations to its use?
- The statistical analysis section is quite detailed, which is commendable. However, some readers might not be familiar with all the statistical methods used. It might be worthwhile to adding a sentence or two explaining what is the rationale for choosing specific tests, expeciually the linear mixed model. As well as, while the authors mention that they used the 'ggplot2', 'rmcorr', 'nlme', 'epitools', and 'emmeans' packages, providing the specific function names would also be helpful.
- Please double-check the consistency of the decimal places used throughout the text.
- In section 2.1, "Their" should be "Their".
- The units for the average milk yield (9197 ± 2036 kg) should be included (i.e., kg/lactation).
Results
- Figure 1 is effective in showing the daily maximum reticulorumen temperature and THI. However, the y-axis label "Rumen temperature maximum (°C)" could be improved for clarity. Perhaps "Maximum Daily Reticulorumen Temperature (°C)" would be more precise.
- In the results section on rumination time, the authors mention that "Daily rumination time decreased throughout the prepartum period..." It would be interesting to see if the authors have any thoughts on why rumination time decreased.
- In Figure 5 and Figure 7, it would be helpful to label the y-axis as "Daily rumination time (min)" and "SCR Heatime raw activity units" for better clarity.
Discussion
- The discussion does a good job of relating the findings to existing literature. However, it might benefit from a more in-depth discussion of the implications of the findings for farm management practices. The authors briefly mention the importance of precision monitoring tools and heat stress management, but they could expand on this a bit more, perhaps by suggesting specific strategies that farmers could implement.
- I recommend discussing the dissemination of your results on dairy cow health during heat stress to the public via social media to address misinformation in animal welfare. Given the prevalence of fake news, especially in animal science, sharing your findings on heat stress effects on dairy cows through social media is crucial. Discuss the potential impact of your research on public understanding. Platforms like Twitter, Facebook, and Instagram can accurately disseminate information beyond academia. Highlighting transparent communication and public engagement through accessible channels will enhance your research's credibility and foster a better-informed society. Use modern tools to combat misinformation and ensure your contributions on dairy cow welfare have a meaningful impact. I would propose adding: "In a parallel vein, a study using Instagram shows social media's effectiveness (10.3168/jds.2024-25347). This study underscores social media's power in conveying complex topics, such as the impact of heat stress on rumination time, reticulorumen temperature, and activity in dairy cows, to a broad audience. Such initiatives complement influencers' role in animal welfare communication by providing tangible examples of how digital platforms can foster community engagement and awareness in specialized areas of animal science."
- The study mentions that 14 out of 40 animals became ill after calving. It would be helpful to discuss what the main health issues were and if there was a way to predict which animals would be affected.
- There are a few instances where the word choice could be more precise. For example, consider replacing "got sick" with "developed health disorders."
Conclusion
- The conclusion provides a good summary of the key findings. However, it could be strengthened by explicitly addressing the limitations of the study. For instance, the authors could acknowledge the relatively small sample size or any specific characteristics of the farm or cows that might limit the generalizability of the results. A brief discussion of how these limitations might be addressed in future research would also be valuable.
- While the conclusion mentions the importance of precision monitoring tools, it could offer more specific recommendations for how these tools can be used to improve dairy cow management. For example, the authors could suggest specific thresholds for rumination time or activity that could trigger interventions by farm staff.
- It states that "Heat stress management with cooling strategies is also crucial for dry cows..." It would be great to add a bit more detail here. What specific cooling strategies do the authors have in mind? Are there any particular best practices that the study supports, report it?
- In the sentence, "Our study confirmed that cows developing postpartum diseases had reduced rumination time already in the prepartum phase...", consider reword "had reduced rumination time already in the prepartum phase" for greater clarity and flow. I suggest: "Our study confirmed that cows developing postpartum diseases already exhibited reduced rumination time in the prepartum phase..."
References
- The list appears to be quite good, which is a positive. but, it's important to double-check that all references cited in the text are present in the list and that the formatting is consistent throughout. I wouldd recommend using a reference management tool to ensure accuracy and consistency such as Mendeley or zotero.
- It would be beneficial to ensure that the references are up-to-date and include the most recent relevant publications in the field. Are there any key papers from the last 2-3 years that should be included to provide the most current context for the research? See my suggestion
- Consider adding DOIs where available to improve the accessibility and discoverability of the cited articles. This is particularly important for online journals and can help readers to easily locate the full text of the references.
- Carefully review the formatting of the references to ensure consistency in terms of author names (e.g., full first name vs. initials), journal titles (e.g., abbreviations), and other bibliographic details. As reported before.
- Double-check for any typos or errors in the references, as these can detract from the overall quality of the manuscript.
Author Response
Rev.: The paper, titled "Assessing the impact of heat stress on rumination time, reticulorumen temperature, and activity during the periparturient period in dairy cows" addresses an important and timely topic in animal science. I found the subject matter of the article absolutley fascinating, and read the manuscript with great interest. The paper aligns well with the scope of the journal, falling as it does into a very relevant area of study. However, I do believe that in its current form, it has several shortcomings that need to be addressed before it can be considered for publication. These issues, which I detail below, range from minor presentational things to some more significant methodological concerns, unfortunatly.
AU: Thank you for your rigorous review and valuable comments and suggestions. We tried to improve the quality of our manuscript accordingly.
Introduction
Rev.: It provides a well overview of the studys context, but it could benefit from clearer articulation of the specific knowledge gap being addressed. But the authors report the importance of early disease detection/identification and the use of PLF tools, the introduction doesn't explicitly state what is lacking in the present literature that this study aims to fill the gap. It would be great to see a sentence or two highlighting what new insights this research is expected to provide. See the suggestions:
Line 58: I suggest citing 10.3389/fvets.2023.1141286 to include more information regarding the longevity and profitability.
Line 60: I suggest citing 10.3389/fvets.2024.1437352 to include more information regarding the metabolic disease.
Line 68: I suggest citing 10.1016/j.vas.2024.100363 to include more information regarding heat stress.
Line 72: I suggest citing 10.3390/ani12162129 regarding the monitoring using boluses for not only temperature but also pH.
Line 81: I suggest citing 10.3390/ani15030458 and 10.3389/fanim.2025.1547395 regarding PLF technologies to monitor rumination time.
AU: Thank you for your suggestions. We revised the text and included the topics and the suggested papers relevant to our study.
Rev.: References reported is quite comprehensive, but at times it feels a bit like a list of facts. To improve the flow and impact, the authors could try to synthesize the existing research more, perhaps by grouping studies that address similar aspects of the topic and by highlighting any conflicting findings or areas of uncertainty. This would help to improve a stronger narrative and make the research question stand out more.
AU: Thank you. We rephrased the text for better clarity.
Rev.: it mentions many factors that influence RT and activity, such as heat stress, calving, and health disorders. but, the relative importance of these factors and how they might interact isn't always clear. It can be helpful if the authors could provide a bit more context, perhaps by explaining which factors are considered most critical in the peri-parturient period and how this study will help to disentangle their effects.
AU: We tried to do our best to improve the clarity and flow of the text.
Rev.: "absolutley" should be "absolutely"
"unfortunatly" should be "unfortunately"
There are a few instances where the phrasing is a bit informal for a scientific paper. For example, "it feels a bit like a list of facts." Consider rephrasing these for a more academic tone.
AU: We did not find the text mentioned above. However, we carefully checked the manuscript and rephrased it where necessary.
Materials and Methods
Rev.: The description of the used animal selection and grouping is generally good, but I have a question about the criteria for defining "clinically healthy" cows as including factor. It would be usefull to have a bit more details on what specific health parameters were assessed and what thresholds were used to classify dairy cows as healthy. Also, were there specific cut-offs used for BCS? Report references thanks
AU: The animals were clinically healthy, meaning that they had no clinical issues (lameness, mastitis, or other diseases) before and at the enrollment. There is no specific cut-off for BCS, as we used BCS only for descriptive statistics. We revised the text.
Rev.: The section on barn microclimate measurements explains how the THI was calculated, which is good. However, it might be worth adding a brief justification for why the THI threshold of 68 was chosen. Is this a widely accepted value in the literature, or is there something specific to this study that warrants its use?
AU: The classic THI thresholds (e.g. 72) were developed for human studies. Since cows are more susceptible to warm weather than humans, it turned out that other THI thresholds should be used. The 68 we used was calculated based on our climate on Hungarian farms, but it was used by other studies, too. It is widely accepted, so we added more references.
Rev.: In the description of the rumination time and activity monitoring, it would be helpful to provide a bit more detail on how the Ruminact HR-Tags work. While the authors mention that the sensor detects the sound of regurgitation, it would be interesting to know more about the specifics of the technology. For example, how accurate and reliable is it? Are there any limitations to its use?
AU: As far as we know from the literature, it is accurate and reliable since there were validation studies (references were mentioned in the manuscript). Unfortunately, we do not know much about the technology, although we contacted the manufacturer for more details. Of course, there are limitations of use, as it was reviewed by Lamanna et al. (2025), one paper you brought to our attention, but these limitations were beyond our study scope. We tried to add some information based on the literature.
Rev.: The statistical analysis section is quite detailed, which is commendable. However, some readers might not be familiar with all the statistical methods used. It might be worthwhile to adding a sentence or two explaining what is the rationale for choosing specific tests, expeciually the linear mixed model. As well as, while the authors mention that they used the 'ggplot2', 'rmcorr', 'nlme', 'epitools', and 'emmeans' packages, providing the specific function names would also be helpful.
AU: The statistical analysis section has been revised. The function names are ggplot(), lme(), rmcorr(), epidate() and emmeans() which seemed an unnecessary repetition of the package names, therefore, we did not include them.
Rev.: Please double-check the consistency of the decimal places used throughout the text.
AU: It was checked, thank you.
Rev: In section 2.1, "Their" should be "Their".
AU: We checked the text for typos and spelling errors.
Rev.: The units for the average milk yield (9197 ± 2036 kg) should be included (i.e., kg/lactation).
AU: The text states that the average milk yield data was calculated for the previous 305 d lactation.
Results
Rev.: Figure 1 is effective in showing the daily maximum reticulorumen temperature and THI. However, the y-axis label "Rumen temperature maximum (°C)" could be improved for clarity. Perhaps "Maximum Daily Reticulorumen Temperature (°C)" would be more precise.
AU: We modified the figures for the more precise labels.
Rev.: In the results section on rumination time, the authors mention that "Daily rumination time decreased throughout the prepartum period..." It would be interesting to see if the authors have any thoughts on why rumination time decreased.
AU: It is a physiological change before calving. We added a reference for this in the discussion.
Rev.: In Figure 5 and Figure 7, it would be helpful to label the y-axis as "Daily rumination time (min)" and "SCR Heatime raw activity units" for better clarity.
AU: We modified the figures for the more precise labels.
Discussion
Rev.: The discussion does a good job of relating the findings to existing literature. However, it might benefit from a more in-depth discussion of the implications of the findings for farm management practices. The authors briefly mention the importance of precision monitoring tools and heat stress management, but they could expand on this a bit more, perhaps by suggesting specific strategies that farmers could implement.
AU: Making predictions would be very useful for the farmers. Our small sample size, however, did not allow us to calculate predictions. This was mentioned in the discussion.
Rev.: I recommend discussing the dissemination of your results on dairy cow health during heat stress to the public via social media to address misinformation in animal welfare. Given the prevalence of fake news, especially in animal science, sharing your findings on heat stress effects on dairy cows through social media is crucial. Discuss the potential impact of your research on public understanding. Platforms like Twitter, Facebook, and Instagram can accurately disseminate information beyond academia. Highlighting transparent communication and public engagement through accessible channels will enhance your research's credibility and foster a better-informed society. Use modern tools to combat misinformation and ensure your contributions on dairy cow welfare have a meaningful impact. I would propose adding: "In a parallel vein, a study using Instagram shows social media's effectiveness (10.3168/jds.2024-25347). This study underscores social media's power in conveying complex topics, such as the impact of heat stress on rumination time, reticulorumen temperature, and activity in dairy cows, to a broad audience. Such initiatives complement influencers' role in animal welfare communication by providing tangible examples of how digital platforms can foster community engagement and awareness in specialized areas of animal science."
AU: Thank you for pointing this out. We also find the dissemination essential, but the paper you suggested adding seems irrelevant to our study.
Rev.: The study mentions that 14 out of 40 animals became ill after calving. It would be helpful to discuss what the main health issues were and if there was a way to predict which animals would be affected.
AU: It is mentioned in the manuscript that a cow was allocated to the diseased group if she had been diagnosed with mastitis, metritis, lameness, or ketosis by the farm veterinarian. In this study, we were curious to see if the measured parameters in otherwise healthy cows before calving could detect changes to help predict which ones would become sick after calving. Such a parameter seems to exist, but our sample size for making the prediction was small. This calculation could be performed with a larger database but only with a similar retrospective grouping. We supplemented the text with these limitations.
Rev.: There are a few instances where the word choice could be more precise. For example, consider replacing "got sick" with "developed health disorders."
AU: We did not find the text mentioned above. However, we carefully checked the text for clearer and more precise wording.
Conclusion
Rev.: The conclusion provides a good summary of the key findings. However, it could be strengthened by explicitly addressing the limitations of the study. For instance, the authors could acknowledge the relatively small sample size or any specific characteristics of the farm or cows that might limit the generalizability of the results. A brief discussion of how these limitations might be addressed in future research would also be valuable.
AU: We believe that conclusions should be focused and concise. Therefore, the limitations were mentioned at the end of the discussion.
Rev.: While the conclusion mentions the importance of precision monitoring tools, it could offer more specific recommendations for how these tools can be used to improve dairy cow management. For example, the authors could suggest specific thresholds for rumination time or activity that could trigger interventions by farm staff.
AU: The conclusions were rephrased to be more specific.
Rev.: It states that "Heat stress management with cooling strategies is also crucial for dry cows..." It would be great to add a bit more detail here. What specific cooling strategies do the authors have in mind? Are there any particular best practices that the study supports, report it?
AU: This part was deleted according to the suggestions by Rev.1.
Rev.: In the sentence, "Our study confirmed that cows developing postpartum diseases had reduced rumination time already in the prepartum phase...", consider reword "had reduced rumination time already in the prepartum phase" for greater clarity and flow. I suggest: "Our study confirmed that cows developing postpartum diseases already exhibited reduced rumination time in the prepartum phase..."
AU: We checked the text for better clarity. The suggested part was also reworded.
References
Rev.: The list appears to be quite good, which is a positive. but, it's important to double-check that all references cited in the text are present in the list and that the formatting is consistent throughout. I wouldd recommend using a reference management tool to ensure accuracy and consistency such as Mendeley or zotero.
AU: We checked all references.
Rev.: It would be beneficial to ensure that the references are up-to-date and include the most recent relevant publications in the field. Are there any key papers from the last 2-3 years that should be included to provide the most current context for the research? See my suggestion
AU: We added new references.
Rev.: Consider adding DOIs where available to improve the accessibility and discoverability of the cited articles. This is particularly important for online journals and can help readers to easily locate the full text of the references.
AU: We added DOIs. However, the MDPI authors' guide does not require adding DOIs.
Rev.: Carefully review the formatting of the references to ensure consistency in terms of author names (e.g., full first name vs. initials), journal titles (e.g., abbreviations), and other bibliographic details. As reported before.
AU: We checked the reference formatting.
Rev.: Double-check for any typos or errors in the references, as these can detract from the overall quality of the manuscript.
AU: We checked the references.
Reviewer 3 Report
Comments and Suggestions for Authors
Simples Summary: in line 21, add how many cows got sick
Abstract:
Line 32 I think that “at risk animals’ is not the most appropriate, maybe change to animal individual
Line 33 - 34. Based on the study's hypothesis, the aim was not only to evaluate but also to compare rumen and behavioral parameters between healthy and sick cows. Review.
Line 38-40 I believe that this brief explanation of the group classification requires improvement. This is because animals that were only diagnosed with disease at the end of the evaluation period may not have exhibited any changes at the beginning, potentially introducing bias when comparing the groups.
line 46 - if it's not significant there's no difference, the two words together are irrelevant
include - statistical analysis
Keywords: change for the words that arent in the title.
Introduction
Line 79. Review – Which technology? City accelerometer that you used in your study.
Line 101 to 104 same comment in the abstract
Change first hypothesis and after the objective.
Material and Methods
Line 122 – 127 the same comment
I believe that this brief explanation of the group classification requires improvement. This is because animals that were only diagnosed with disease at the end of the evaluation period may not have exhibited any changes at the beginning, potentially introducing bias when comparing the groups.
Review Statistical analysis: Add mathematical models to the statistical analysis. It will help you better understand how the data was analyzed.
Results:
Figure 6 : Explain why this analysis is only carried out for the rumination time parameter and not for the others.
General comments:
All citations should change ( ) for [ ] as showed in template.
Author Response
Rev.: Simples Summary: in line 21, add how many cows got sick
AU: We added how many were healthy or ill.
Rev.: Line 32 I think that "at risk animals' is not the most appropriate, maybe change to animal individual
AU: We rephrased this part.
Rev.: Line 33 - 34. Based on the study's hypothesis, the aim was not only to evaluate but also to compare rumen and behavioral parameters between healthy and sick cows. Review.
AU: Yes, indeed. It is written a sentence later.
Rev.: Line 38-40 I believe that this brief explanation of the group classification requires improvement. This is because animals that were only diagnosed with disease at the end of the evaluation period may not have exhibited any changes at the beginning, potentially introducing bias when comparing the groups.
AU: It was impossible to predict at the enrollment which otherwise healthy cows would become sick after calving, so the retrospective assignment of groups was necessary and, indeed, the only possible way. We were curious to see if the measured parameters in otherwise healthy cows could detect changes to help predict which ones would become sick after calving. Such a parameter seems to exist, but our sample size for making the prediction was small. This calculation could be performed with a larger database but only with a similar retrospective grouping. The evaluation was not performed at the end, and the authors did not even perform it. The farm veterinarian diagnosed the postpartum problems and allocated the cows into the groups. The abstract word number is limited, so we added some explanations to the materials, methods, and discussion.
Rev.: line 46 - if it's not significant there's no difference, the two words together are irrelevant
AU: We deleted the word significant.
include - statistical analysis
AU: The abstract word number is limited, so there is no place to detail the statistical methods.
Rev: Keywords: change for the words that arent in the title.
AU: We revised the keywords. Also, we changed the title to one better fit to our study.
Rev.: Line 79. Review – Which technology? City accelerometer that you used in your study.
AU: Citation was added.
Rev.: Line 101 to 104 same comment in the abstract
AU: The text is supplemented accordingly.
Rev.: Change first hypothesis and after the objective.
AU: We changed the text.
Rev.: Line 122 – 127 the same comment. I believe that this brief explanation of the group classification requires improvement. This is because animals that were only diagnosed with disease at the end of the evaluation period may not have exhibited any changes at the beginning, potentially introducing bias when comparing the groups.
AU: The answer is the same as above ?
Rev.: Review Statistical analysis: Add mathematical models to the statistical analysis. It will help you better understand how the data was analyzed.
AU: The statistical analysis section has been revised. The model formula has now been included.
Rev.: Figure 6 : Explain why this analysis is only carried out for the rumination time parameter and not for the others.
AU: Since there was no significant difference in the rumination time between the two groups just before and after calving, it was assumed that this was due to the differing degrees of change before calving. Therefore, we analyzed this close peripartum period (-1 to 2 DIM, indicated with a blue background in Figure 5). Figure 6 displays the results of this separate analysis, showing a significant difference between the two groups on days -1 and 2. There was no such difference between the groups in the other parameters, so this separate analysis was unnecessary.
Rev.: All citations should change ( ) for [ ] as showed in template.
AU: The text is checked for this.
Round 2
Reviewer 1 Report
Comments and Suggestions for Authors
The manuscript presents a relevant and well-executed study that investigates the physiological responses of dairy cows to heat stress during the periparturient period. The topic is of high relevance to animal science and veterinary medicine, especially in the context of climate change and its effects on animal welfare and productivity.
The revised version of the manuscript adequately addresses the issues raised in the previous round of review. In light of these revisions, the article now meets the standards of scientific quality and clarity required for publication in Animals.
Author Response
Rev.: The manuscript presents a relevant and well-executed study that investigates the physiological responses of dairy cows to heat stress during the periparturient period. The topic is of high relevance to animal science and veterinary medicine, especially in the context of climate change and its effects on animal welfare and productivity.
The revised version of the manuscript adequately addresses the issues raised in the previous round of review. In light of these revisions, the article now meets the standards of scientific quality and clarity required for publication in Animals.
AU: Thank you for your review.
Reviewer 3 Report
Comments and Suggestions for Authors
Even with the inclusion, I believe that the use of the equation tool can improve the inclusion of the models in the work.
Another point about replacing () with [] in the references, I don't understand why it wasn't accepted.
Author Response
Rev.: Even with the inclusion, I believe that the use of the equation tool can improve the inclusion of the models in the work.
Another point about replacing () with [] in the references, I don't understand why it wasn't accepted.
AU: We prepared the equations with the equation tool of Word. Regarding the [], it is not that we did not accept it, but we simply forgot about it among the many other things to improve.